# Ten Years (2011–2021) of the Italian Lombardy ADHD Register for the Diagnosis and Treatment of Children and Adolescents with ADHD

**DOI:** 10.3390/children8070598

**Published:** 2021-07-15

**Authors:** Maurizio Bonati, Francesca Scarpellini, Massimo Cartabia, Michele Zanetti

**Affiliations:** Laboratory for Mother and Child Health, Department of Public Health, Istituto di Ricerche Farmacologiche Mario Negri IRCCS, 20156 Milan, Italy; francesca.scarpellini@marionegri.it (F.S.); massimo.cartabia@marionegri.it (M.C.); michele.zanetti@marionegri.it (M.Z.)

**Keywords:** attention deficit disorder with hyperactivity, child, adolescent, mental health, chronic disease, register, clinical protocol

## Abstract

Background: The purpose of this article is to update the diagnostic assessment, therapeutic approach, and 12–18 month follow-up of patients added to the Italian Lombardy Attention Deficit Hyperactivity Disorder (ADHD) Register. Methods: Medical records of patients added to the Registry from 2011 to 2021 were analysed. Results: 4091 of 5934 patients met the criteria for a diagnosis of ADHD, and 20.3% of them presented a familiarity with the disorder. A total of 2879 children (70.4%) had at least one comorbidity disorder, in prevalence a learning disorder (39%). Nearly all (95.9%) received at least one psychological prescription, 17.9% of them almost one pharmacological treatment, and 15.6% a combination of both. Values of ≥5 of the Clinical Global Impression—Severity scale (CGI-S) are more commonly presented by patients with a pharmacological prescription than with a psychological treatment (*p* < 0.0001). A significant improvement was reported in half of the patients followed after 1 year, with Clinical Global Impression—Improvement scale (CGI-I) ≤ 3. In all, 233 of 4091 are 18-year-old patients. Conclusions: A ten-year systematic monitoring of models of care was a fruitful shared and collaborative initiative in order to promote significant improvement in clinical practice, providing effective and continuous quality of care. The unique experience reported here should spread.

## 1. Introduction

Attention deficit hyperactivity disorder (ADHD) is a neurodevelopmental disorder that affects 5.9% of children and persists into adulthood for two-thirds of them [1,2], with great impairments in academic achievement and work [3]. The core symptoms are inattention, restlessness and impulsivity, which are more frequent in boys than girls (ratio 3:1). In Italy, the prevalence of the disorder ranges from 1.1 to 3.1% of the paediatric population, considering only subjects with a diagnosis confirmed by clinical evaluation [4].

The wide variability depends on the different diagnostic procedures adopted to assess children and the criteria used, and the period of time over which assessment is conducted [5]. The peak age of diagnosis of ADHD is in primary school children aged 5–10 years [6], and children born later in the school year are more likely to receive an ADHD diagnosis than their same school-year peers [7]. According to the national and international guidelines [8,9], the diagnosis of ADHD is based on a careful and systematic assessment of a lifetime history of symptoms, childhood onset, and impairment in some contexts (schools, relationships, home) [10]. Information about the medical history of psychiatric and neurological problems is also important. Psychiatric comorbidity is thus a clinically important factor that contributes to the persistence of ADHD in adulthood [11]. Oppositional defiant disorder (ODD), conduct disorder (CD) and autism spectrum disorder are the most common conditions associated with ADHD [12]. Concerning treatment, the guidelines suggest using a multimodal treatment combining psychosocial interventions with pharmacological therapies. Psychological therapies involve parents and teachers with training and a range of cognitive behavioural approaches to the patient. Medication includes stimulants, in particular methylphenidate, as a first choice and the most effective therapy. The stimulant medications for ADHD are more effective than non-stimulant medications but are also more likely to be diverted, misused, and abused [13,14]. Otherwise, non-medication treatments for ADHD are less effective than medication treatments for ADHD symptoms, but are frequently useful to help problems that remain after medication has been optimised [1]. A recent meta-analysis of the literature highlighted the positive effect of psychological interventions on ADHD cognitive symptomatology and supports the inclusion of non-pharmacological interventions in conjunction with the commonly used pharmacological treatments [15]. Despite the existence of clear and specific guidelines, access to services is limited [16,17], the waiting times for diagnosis are too long [18], and the treatment outcomes depend on many factors, such as the presence of comorbidities [19]. The Regional ADHD Registry was activated in June 2011 with the purpose of collecting data about the diagnosis and treatment of ADHD patients who had access to the 18 centres, with particular attention to the monitoring of pharmacological treatment. The Regional Registry was part of a more general project aiming to ensure appropriate ADHD management for children and adolescents once the disorder is suspected; the data recorded include common assessment processes as well as psychoeducational interventions for healthcare workers of the Lombardy region healthcare system [2]. After 10 years of the project, we aimed to update the diagnostic assessment and therapeutic approaches proposed to 5–17-year-old youths who had access to any of the 18 ADHD reference centres of the Lombardy region. In particular, we analysed the clinical characteristics of ADHD patients and the relation with the treatment prescription and the elevation in some scales of the test used for the diagnosis. We explored if there was an improvement on their Clinical Global Impression—Improvement scale (CGI-I) scores after 12–18 months of follow-up.

## 2. Materials and Methods

A retrospective study based on medical records was conducted. Data were identified from the Regional ADHD Registry. Formal ethical review board approval was not required for the present updating because it was previously approved by the Institutional Review Board of the Istituto di Ricerche Farmacologiche Mario Negri IRCCS, Milan, Italy. Written informed consent was obtained from all patients before data collection. We used the previously described methodology and reported data concerning the local health setting [7], the characteristics of the ADHD Registry activated in Lombardy in June 2011 [20,21], the systematic work carried out by the 18 ADHD centres [19], and the diagnostic assessment and the treatment conducted by all involved clinicians, according to the national and international guidelines [8,9]. Behavioural and emotional problems were highlighted with the most used and validated rating scales for parents and teachers, Conners’ Parent Rating Scale (CPRS) [22], Conners’ Teacher Rating Scale (CTRS) [23], and the Child Behaviour Checklist (CBCL) [24], while symptom severity and symptom improvement were quantified, respectively, with the use of the Clinical Global Impression—Severity scale and the Clinical Global Impressions—Improvement scale [25]. Results from the scales were analysed and compared with the perceptions of parents and teachers, as well as the perceptions of mothers and fathers. The Clinical Global Impressions—Improvement Scale scores were analysed after 12–18 months of follow-up. The percentages of completeness of the seven areas of the shared diagnostic assessment (Clinical Interview, Neurological Examination, IQ Evaluation, Diagnostic Interview, Parents and Teachers Assessment, Clinical Severity Evaluation) of all regional centres were analysed and displayed on radar chart axes with a range of 0 to 100%

Data were extracted from the database and analyses were updated on 1 April 2021, and data referred to patients added between 2011 and 2021.

### Data Analyses

All data were entered in an SAS/STAT database (SAS Version 9, SAS Institute, Inc., Cary, NC, USA). Descriptive statistics were computed for the entire study population and for subgroups. The Wilcoxon test was used to compare continuous variables, whereas chi-square tests were used to compare categorical variables. V-Cramer and Wilcoxon effect sizes were calculated (Appendix A). Both values vary from 0 to 1; the closer the value was to 1, the stronger the significant difference between the categorical and the continuous variables was. A multivariate logistic regression analysis with stepwise selection was also carried out to assess the determinants of disease and treatment. Moreover, interrater agreement (parents vs. teachers; mothers vs. fathers) on symptom scores for each diagnostic scale was established by Kappa coefficient of agreement (K). The results are presented as the number, frequency (%), and mean or median; *p* < 0.05 was considered to be significant.

## 3. Results

A total of 7053 children were added to the registry from June 2011 to December 2021, of whom 6188 were children and adolescents accessing the ADHD centres for the first time (range 89–1010 patients per centre, median = 248) for suspected ADHD diagnosis. Most of the patients (5934) had completed the diagnostic assessment (Table 1). Children had a median age of 9 years (range 7–11); most of them were males (4960 (83.6%)) and 974 (16.4%) were females. In all, 4091 patients received a diagnosis of ADHD based on the Diagnostic and Statistical Manual of Mental Disorders [26] criteria, 3484 (85.2%) of whom were males and 607 (14.8%) females. The cumulative incidence of ADHD in the 2011 and 2021 period was valued to be 0.26% (95% confidence interval (CI = [0.94–1.24]) of the resident population of the same age range, with a spike at 8 years of age (Figure 1).

The characteristics strongly associated with ADHD were lower age, male gender, only child, not born in Italy, adopted, support teacher, lower educational level of parents, unemployed father, ADHD familiarity and psychiatric comorbidity (Table 1). According to both univariate and multivariate analysis, only child (odds ratio (OR) = 1.22, 95% CI = [1.07–1.41]), adopted (OR = 1.64, 95% CI = [1.07–2.52]), primary school (OR = 1.15, 95% CI = [1.00–1.31]), support teacher (OR = 2.82, 95% CI = [2.19–3.64]), employed mother (OR = 1.14, 95% CI = [1.01–1.20]), unemployed father (OR = 1.31, 95% CI = [1.08–1.58]), and ADHD familiarity (OR = 2.21, 95% CI = [1.86–2.63]) were higher in ADHD patients. In all, 2879 of 3956 (70.4%) had at least one psychiatric disorder (1079 without ADHD), whereas 387 (6.5%) had another comorbidity chronic disease. Learning disorders (OR = 1.37, 95% CI = [1.21–1.55]), sleep disorders (OR = 1.83, 95% CI = [1.50–2.23]), oppositional defiant disorder (ODD) (OR = 2.87, 95% CI = [2.27–3.64]), language disorders (OR = 1.36, 95% CI = [1.05–1.77]), tics (OR = 1.87, 95% CI = [1.16–3.03]) and motor coordination disorders (OR = 1.68, 95% CI = [1.04–2.72]) were higher in ADHD patients. Anxiety (OR = 0.70, 95% CI = [0.56–0.87]) and conduct disorder (OR = 0.49, 95% CI = [0.32–0.76]) were higher in patients without ADHD. The presence of a neurological condition was more frequent (*n* = 121, 2%).

### 3.1. Prescription after Diagnosis

In all, 4016 ADHD patients (98.2%) received at least one prescription: 3282 (80.2%) received only psychological treatment, 94 (2.3%) only pharmacotherapy and 640 patients (15.6%) received both pharmacological and psychological treatment. A total of 734 patients received a drug prescription, 679 (16.6%) of them methylphenidate (0.5–80 mg daily), of those 21 (0.5%) together with another drug (i.e., risperidone, aripiprazole, haloperidol, lorazepam, sertraline, alprazolam, fluvoxamine, clomipramine, delorazepam), 23 (0.6%) atomoxetine (10–60 mg daily), and 53 (1.3%) other psychotropic drugs. Among the 3922 (95.9%) patients prescribed psychoeducational treatment, 2631 (64.3%) received at least one type of training intervention (parents, teacher or child), whereas 2820 (68.9%) received other psychological treatment. Parent training was the most frequently proposed psychological treatment (*n* = 2311, 56.5%), followed by child training (*n* = 1556, 38%) and teacher training (*n* = 870, 21.3%). All training types were proposed to 519 (12.7%) patients. In all, 2485 (61.9%) patients had an ADHD Combined subtype (ADHD-C), whereas 1218 (30.3%) were diagnosed with Inattentive Type (ADHD-I) and 313 (7.8%) Hyperactive Type (ADHD-HI) subtypes. Of all 4091 patients diagnosed with ADHD, 4016 patients received a prescription (Table 2).

Those with an ADHD-C diagnosis were treated more commonly with drug therapy (79.8%) than with psychological treatment (57.9%), *p* < 0.0001. Otherwise, ADHD-I patients were more often treated with psychological treatment (33.4%) than drug therapy (16.6%), *p* < 0.0001. Univariate analysis between patients treated with pharmacological and psychological treatment highlighted several significant differences, as reported in Table 2. Multivariate analysis highlighted a higher probability to receive both medication and psychological prescription for those children who received a diagnosis of the ADHD combination type (OR = 3.48, 95% CI = [1.84–6.59]), a pathological score on the Hyperactivity Scales of (OR = 1.48, 95% CI [1.10–1.99]) or on the ADHD Index of CPRS (OR = 2.79, 95% CI = [1.82–4.28]), a Clinical Global Impression—Severity scale (CGI-S) score of 5 or above (OR = 8.04, 95% CI = [6.35–10.19]), and a comorbidity condition, such as ODD (OR = 1.64, 95% CI = [1.25–2.15]), cognitive delay (OR = 2.47, 95% CI = [1.65–3.69]), tics (OR = 3.60, 95% CI = [2.02–6.43]) and coordination disorders (OR = 3.67, 95% CI = [2.02–6.67]).

### 3.2. Parents and Teachers Rating Scales

Results from the rating scales, filled out by the parents and teachers, were compared in order to highlight differences or similarities about how adults perceived the symptom severity and behavioural problems of the child. The same comparison was made between mothers’ and fathers’ perception (Table 3).

The results highlighted that parents tended to perceive more cognitive problems (CPRS-I = 3270, 66.6%; CTRS-I = 2682, 54.6%; *p* < 0.0001) than teachers, who reported more behavioural (CTRS-H = 2848, 58%; CPRS-H = 2646, 53.9%; *p* < 0.0001) and emotional problems (CTRS-E = 1942, 39.6%; CPRS-E = 1645, 33.5%; *p* < 0.0001). Mothers reported more child pathological problems than fathers did on both subscales (CBCL-I = 223, 20.6%; CBC-I father = 159, 14.7%; *p* = 0.0003) (CBCL-E = 254, 23.5%; CBCL-E father = 204, 18.9%; *p* = 0.0085) and on the CBCL-Total score (CBCL-T = 349, 32.3%; CBCL-T father = 255, 23.6%; *p* < 0.0001). Despite these differences, kappa values highlighted a “fair” agreement between parents and teachers (>60%) and “excellent” agreement between mothers and fathers (>85%).

### 3.3. Continuity of Care and Management

Throughout the regional database system, data on patient care from the first access to the diagnosis and data on treatment prescriptions and follow-up visits were systematically collected. As shown in Figure 2, the diagnostic evaluation was full and accurate: each axis has a range of 0 to 100% and represents one of the seven areas of the shared diagnostic assessment (Clinical Interview, Neurological Examination, IQ Evaluation, Diagnostic Interview, Parents and Teachers Assessment, Clinical Severity Evaluation), while the three datasets represent the performance scores of the most (average = 100%) and least compliant (average = 91.01%) ADHD centre, as well as the total completeness (average = 97.86%) estimated by the analysis of data recorded by all 18 ADHD centres. Overall, 320 of 4091 patients with a diagnosis of ADHD were discharged during the first 3 months; 1468 patients with ADHD had been monitored for more than 1 year after the diagnosis, half of whom had a significant improvement with CGI-I scores of 1–3, and the majority of these (89%) were in a stable condition with scores of 4 on the CGI-I. In all, 755 patients reached the legal age (range 18–27 years), 31.3% of whom just turned 18 years old.

## 4. Discussion

Ten years after the creation of the Lombardy Registry Project, clinical and service assessment data revealed the effectiveness and usefulness of this regional project in providing assistance and continuity of care to ADHD patients and their families. Over the years, the registry was monitored to achieve clinical improvement, using systematic activities and an interactive system evaluation to test the change, according to the main clinical quality improvement features [27]. The clinical characteristics of the ADHD patients of the Lombardy Registry Project were in line with the literature; the peak age of diagnosis was in primary school children aged 5–10 years [6], and LD was the main psychiatric comorbidity followed by ODD, anxiety, and sleeping disorders [1,12]. The most associated chronic disease was a neurological condition. Concerning treatment, data extracted from the registry highlighted a relation between some clinical characteristics and the type of prescription at diagnosis; according to the literature, the symptom severity increased the likelihood of being prescribed ADHD medication [28]. The higher the CGI-S score, the higher the probability of receiving a medical prescription, in particular for patients with ODD, an intellectual disability, tics or a coordination disorder. Differently from what we expected, learning problems were not associated with being prescribed medication, suggesting that learning problems may not be pertinent to pharmacological treatment decisions for children with ADHD. Pharmacological prescription was infrequent (18%), and nearly all of the patients (96%) received a psychological prescription such as child, parent or teacher training. Comparing data with higher rates reported in other countries [29], in Italy child psychiatrists’ professional attitude leaned more toward behavioural treatments than to the use of drugs [30]. In general, half of the patients with a diagnosis of ADHD and in treatment for almost one year (follow-up between 12 and 18 months) reported an improvement in their level of symptom severity on the CGI-I score. These data were comforting, suggesting the clinical care utility of a continuously monitored, standardised system. The project represents a great opportunity to improve collaboration, share assessment approaches and promote the continuity of care of patients affected by ADHD, monitoring their treatments and healthcare pathway. This represented an important value for the project; continuously monitoring and sharing data is an important approach in order to ensure the quality of care. The ADHD project could represent an example of a healthcare system for other chronic conditions and psychiatric disorders in childhood in order to promote the continuity and improvement of childcare. Particular attention should be paid to a particular phase called transition—the passage between child care to adult care; after ten years of the project, many children became adolescents, and some of them are near the boundary age. The transition process not only concerns healthcare but also involves the transition to adulthood, finding employment or continuing the education process; therefore, it is important for the adolescent to be prepared to manage their medical condition and pharmacological treatment. Once reaching adulthood, the risk is being discharged by the services and not receiving prescribed medication. In order to avoid this situation, it is very important to promote the continuity and monitoring of childcare. A great deal has been done, but a lot of work is still necessary for the best management of ADHD across the lifespan. The limit of the project’s approach is that it represents a national uniqueness. Moreover, the registry, due to its nature of being an observatory of healthcare provided by a service (even if public), does not contemplate the rate of patients who interrupted the care pathway or who did not start it, failing to show up from the start. Despite the remarkable improvements made in the last twenty years for providing appropriate diagnostic and treatment services for children with ADHD, also as a result of the landmark Multimodal Treatment Study of Children with Attention Deficit/Hyperactivity Disorder (MTA) trial [31], practice is still different between and within countries. Recommendations and guidelines for ADHD management in children and adolescents were produced by several parties and individual centres adapted their care. The findings reported here are not different from others reported in the literature where health services and care are different. However, the heart of the Italian Lombardy ADHD Registry lies in the approach: collaborative, shared between and within the participating centres, over time. It is an unusual approach in the interest of the patients and carers.

## 5. Conclusions

The Regional ADHD Registry represents a distinctive tool, a unique experience in the international context, to help guarantee a shared pathway of care in ADHD children. Continuous, systematic monitoring allows resources to be invested appropriately, such as in promoting progressive and significant improvement in clinical practice, ensuring a shared and efficient quality of care. Training initiatives involving clinicians, patients, parents and teachers may be useful in order to raise awareness about the disorder in clinical practice.

## Figures and Tables

**Figure 1 children-08-00598-f001:**
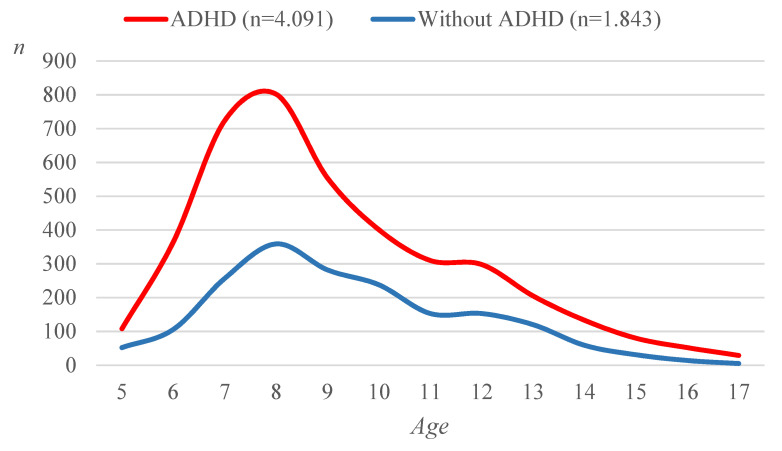
Age of the ADHD patients.

**Figure 2 children-08-00598-f002:**
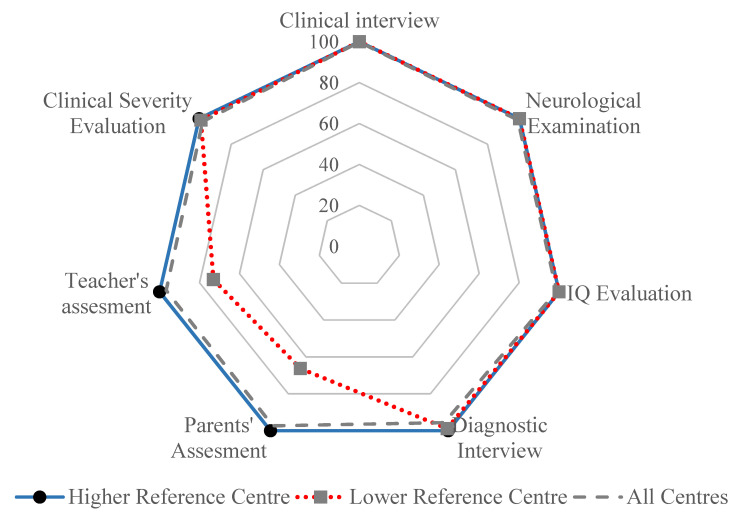
Rate of completeness levels based on data inputted in the registry.

**Table 1 children-08-00598-t001:** Demographic characteristics of the ADHD patients.

		ADHD Yes	ADHD No	Total	*p*	OR (IC 95%)	Logistic
Children		4091	1843	5934			
Age at diagnosis	median (q1–q3)	9.0 (7.0–11.0)	9.0 (8.0–11.0)	9.0 (7.0–11.0)	<0.0001 *		
	media (ds)	9.2 (2.6)	9.4 (2.4)	9.2 (2.5)			
	(min–max)	(5.0–17.0)	(5.0–17.0)	(5.0–17.0)			
	Missing	30	14	44			
School age at diagnosis	5–11	3.264 (80.4)	1.447 (79.1)	4.711 (80.0)	0.2635	1.08 (0.94–1.24)	
	12–17	797 (19.6)	382 (20.9)	1.179 (20.0)		1.00 (Ref.)	
	Missing	30	14	44			
Gender	Female	607 (14.8)	367 (19.9)	974 (16.4)	<0.0001 *	1.00 (Ref.)	
	Male	3.484 (85.2)	1.476 (80.1)	4.960 (83.6)		1.43 (1.24–1.65) *	1.36 (1.17–1.58)
	Missing	-	-	-			
Only child	Yes	1.054 (25.8)	396 (21.5)	1.450 (24.5)	0.0004	1.27 (1.11–1.45)	1.22 (1.07–1.41)
	No	3.028 (74.2)	1.443 (78.5)	4.471 (75.5)		1.00 (Ref.)	
	Missing	9	4	13			
Born in Italy	Yes	3.869 (94.6)	1.774 (96.3)	5.643 (95.1)	0.0067	1.00 (Ref.)	
	No	220 (5.4)	69 (3.7)	289 (4.9)		1.46 (1.11–1.93)	
	Missing	2	-	2			
Adopted	Yes	149 (3.6)	34 (1.8)	183 (3.1)	0.0002	2.01 (1.38–2.93)	1.64 (1.07–2.52)
	No	3.938 (96.4)	1.807 (98.2)	5.745 (96.9)		1.00 (Ref.)	
	Missing	4	2	6			
School	Primary School	3.124 (76.4)	1.371 (74.5)	4.495 (75.8)	0.1124	1.11 (0.98–1.26)	1.15 (1.00–1.31)
	Middle/High School	964 (23.6)	469 (25.5)	1.433 (24.2)		1.00 (Ref.)	
	Missing	3	3	6			
Support teacher	Yes	514 (12.6)	79 (4.3)	593 (10.0)	<0.0001	3.21 (2.51–4.09)	2.82 (2.19–3.64)
	No	3.577 (87.4)	1.764 (95.7)	5.341 (90.0)		1.00 (Ref.)	
	Missing	-	-	-			
Educational level of mother	Yes	2.313 (56.5)	1.116 (60.6)	3.429 (57.8)	0.0038	0.85 (0.76–0.95)	
	No	1.778 (43.5)	727 (39.4)	2.505 (42.2)		1.00 (Ref.)	
	Missing	-	-	-			
Educational level of father	Yes	1.865 (45.6)	934 (50.7)	2.799 (47.2)	0.0003	1.00 (Ref.)	
	No	2.226 (54.4)	909 (49.3)	3.135 (52.8)		1.23 (1.10–1.37)	
	Missing	-	-	-			
Mother employed	Yes	2.729 (66.7)	1.227 (66.6)	3.956 (66.7)	0.9210	1.01 (0.90–1.13)	1.14 (1.01–1.30)
	No	1.362 (33.3)	616 (33.4)	1.978 (33.3)		1.00 (Ref.)	
	Missing	-	-	-			
Father employed	Yes	3.411 (83.4)	1.624 (88.1)	5.035 (84.9)	<0.0001	1.00 (Ref.)	
	No	680 (16.6)	219 (11.9)	899 (15.1)		1.48 (1.26–1.74)	1.31 (1.08–1.58)
	Missing	-	-	-			
ADHD familiarity	Yes	831 (20.3)	193 (10.5)	1.024 (17.3)	<0.0001	2.18 (1.84–2.58)	2.21 (1.86–2.63)
	No	3.260 (79.7)	1.650 (89.5)	4.910 (82.7)		1.00 (Ref.)	
	Missing	-	-	-			
Psychiatric comorbidity	Yes	2.879 (70.4)	1.079 (58.5)	3.958 (66.7)	<0.0001	1.68 (1.50–1.89)	
	No	1.212 (29.6)	764 (41.5)	1.976 (33.3)		1.00 (Ref.)	
	Missing	-	-	-			
Type of comorbidity (flag)	Learning disorder	1.594 (39.0)	613 (33.3)	2.207 (37.2)	<0.0001	1.28 (1.14–1.44)	1.37 (1.21–1.55)
	Sleeping disorder	582 (14.2)	145 (7.9)	727 (12.3)	<0.0001	1.94 (1.60–2.35)	1.83 (1.50–2.23)
	ODD	569 (13.9)	93 (5.0)	662 (11.2)	<0.0001	3.04 (2.42–3.81)	2.87 (2.27–3.64)
	Anxiety	280 (6.8)	162 (8.8)	442 (7.4)	0.0083	0.76 (0.62–0.93)	0.70 (0.56–0.87)
	Language disorder	287 (7.0)	88 (4.8)	375 (6.3)	0.0010	1.50 (1.18–1.92)	1.36 (1.05–1.77)
	Tic	95 (2.3)	22 (1.2)	117 (2.0)	0.0038	1.97 (1.23–3.14)	1.87 (1.16–3.03)
	Conduct disorder	69 (1.7)	41 (2.2)	110 (1.9)	0.1551	0.75 (0.51–1.11)	0.49 (0.32–0.76)
	Coordination disorder	95 (2.3)	23 (1.2)	118 (2.0)	0.0061	1.88 (1.19–2.98)	1.68 (1.04–2.72)
Chronic disease	Yes	265 (6.5)	122 (6.6)	387 (6.5)	0.8376	0.98 (0.78–1.22)	
	No	3.826 (93.5)	1.721 (93.4)	5.547 (93.5)		1.00 (Ref.)	
	Missing	-	-	-			
Type of chronic disease (flag)	Neurological	91 (2.2)	30 (1.6)	121 (2.0)	0.1324	1.37 (0.91–2.08)	
	Breathing	60 (1.5)	32 (1.7)	92 (1.6)	0.4365	0.84 (0.55–1.30)	
	Gastrointestinal	18 (0.4)	11 (0.6)	29 (0.5)	0.4227	0.74 (0.35–1.56)	

ADHD = attention deficit hyperactivity disorder, ODD = oppositional defiant disorder, *p* = chi-squared test (categorical variables) or Wilcoxon test (continuous variables), OR = odds ratio, CI = confidence interval, Logistic = OR and CI from the multivariate logistic regression model with stepwise type selection. (*) = *p* < 0.05.

**Table 2 children-08-00598-t002:** Clinical characteristics of the ADHD patients by treatment prescription.

		Pharmacological Treatment	Psychological Treatment	Total	*p*	OR (IC 95%)	Logistic
		734	3282	4016			
ADHD Subtype	Combined	586 (79.8)	1.899 (57.9)	2.485 (61.9)	<0.0001 *	3.41 (2.26–5.14)	* 3.48 (1.84–6.59)
	Inattentive	122 (16.6)	1.096 (33.4)	1.218 (30.3)		1.23 (0.79–1.91)	2.34 (1.19–4.62)
	Hyperactive	26 (3.5)	287 (8.7)	313 (7.8)		1.00 (Ref.)	
	Missing	-	-	-			
QI pathologic	Yes	82 (11.4)	79 (2.4)	161 (4.1)	<0.0001 *	5.18 (3.76–7.13)	
	No	635 (88.6)	3.166 (97.6)	3.801 (95.9)		1.00 (Ref.)	
	Missing	17	37	54			
CPRS-O	Pathological	384 (63.6)	1.305 (42.6)	1.689 (46.0)	<0.0001 *	2.36 (1.97–2.82)	
	Normal	220 (36.4)	1.761 (57.4)	1.981 (54.0)		1.00 (Ref.)	
	Missing	130	216	346			
CTRS-O	Pathological	310 (59.0)	1.219 (41.7)	1.529 (44.3)	<0.0001 *	2.02 (1.67–2.44)	
	Normal	215 (41.0)	1.707 (58.3)	1.922 (55.7)		1.00 (Ref.)	
	Missing	209	356	565			
CPRS-I	Pathological	515 (85.3)	2.176 (70.9)	2.691 (73.3)	<0.0001 *	2.37 (1.87–3.01)	
	Normal	89 (14.7)	892 (29.1)	981 (26.7)		1.00 (Ref.)	
	Missing	130	214	344			
CTRS-I	Pathological	362 (69.0)	1.717 (58.6)	2.079 (60.2)	<0.0001 *	1.57 (1.28–1.91)	
	Normal	163 (31.0)	1.211 (41.4)	1.374 (39.8)		1.00 (Ref.)	
	Missing	209	354	563			
CPRS-H	Pathological	477 (79.0)	1.840 (60.0)	2.317 (63.1)	<0.0001 *	2.51 (2.03–3.09)	* 1.48 (1.10–1.99)
	Normal	127 (21.0)	1.228 (40.0)	1.355 (36.9)		1.00 (Ref.)	
	Missing	130	214	344			
CTRS-H	Pathological	403 (76.6)	1.893 (64.7)	2.296 (66.5)	<0.0001 *	1.79 (1.44–2.22)	
	Normal	123 (23.4)	1.035 (35.3)	1.158 (33.5)		1.00 (Ref.)	
	Missing	208	354	562			
CPRS-ADHD	Pathological	557 (92.2)	2.371 (77.3)	2.928 (79.7)	<0.0001 *	3.48 (2.56–4.75)	* 2.79 (1.82–4.28)
	Normal	47 (7.8)	697 (22.7)	744 (20.3)		1.00 (Ref.)	
	Missing	130	214	344			
CTRS-ADHD	Pathological	459 (87.3)	2.285 (78.1)	2.744 (79.5)	<0.0001 *	1.92 (1.47–2.52)	
	Normal	67 (12.7)	642 (21.9)	709 (20.5)		1.00 (Ref.)	
	Missing	208	355	563			
CGI-S	5–7	516 (71.9)	597 (18.8)	1.113 (28.6)	<0.0001 *	11.02 (9.15–13.26) *	8.04 (6.35–10.19)
	1–4	202 (28.1)	2.575 (81.2)	2.777 (71.4)		1.00 (Ref.)	
	Missing	16	110	126			
Psychiatric comorbidity	Yes	612 (83.4)	2.215 (67.5)	2.827 (70.4)	<0.0001 *	2.42 (1.96–2.97) *	
	No	122 (16.6)	1.067 (32.5)	1.189 (29.6)		1.00 (Ref.)	
	Missing	-	-	-			
Type of comorbidity (flag)	Learning disorder	265 (36.1)	1.296 (39.5)	1.561 (38.9)	0.0890	0.87 (0.73–1.02)	
	Sleeping disorder	130 (17.7)	443 (13.5)	573 (14.3)	0.0032 *	1.38 (1.11–1.71)	
	ODD	209 (28.5)	356 (10.8)	565 (14.1)	<0.0001 *	3.27 (2.69–3.97)	1.64 (1.25–2.15)
	Anxiety	70 (9.5)	208 (6.3)	278 (6.9)	0.0020 *	1.56 (1.17–2.07)	
	Intellectual disability	94 (12.8)	150 (4.6)	244 (6.1)	<0.0001 *	3.07 (2.34–4.02)	2.47 (1.65–3.69)
	Mood disorder	57 (7.8)	170 (5.2)	227 (5.7)	0.0061 *	1.54 (1.13–2.11)	
	Language disorder	61 (8.3)	221 (6.7)	282 (7.0)	0.1307	1.26 (0.93–1.69)	
	Tic	40 (5.4)	54 (1.6)	94 (2.3)	<0.0001 *	3.45 (2.27–5.23)	3.60 (2.02–6.43)
	Conduct disorder	29 (4.0)	39 (1.2)	68 (1.7)	<0.0001 *	3.42 (2.10–5.57)	
	Autism	60 (8.2)	65 (2.0)	125 (3.1)	<0.0001 *	4.41 (3.07–6.32)	
	Coordination disorder	28 (3.8)	63 (1.9)	91 (2.3)	0.0018 *	2.03 (1.29–3.19)	3.67 (2.02–6.67)
	Other	17 (2.3)	66 (2.0)	83 (2.1)	0.5994	1.16 (0.67–1.98)	

CPRS = Conners’ Parent Rating Scale (CPRS-O = Oppositive Scale; CPRS-I = Inattention Scale; CPRS-H = Hyperactivity Scale; CPRS-ADHD = ADHD Index; CPRS-E = Emotion Lability Scale); CTRS = Conners’ Teacher Rating Scale (CTRS-O = Oppositive Scale; CTRS-I = Inattention Scale; CTRS-H = Hyperactivity Scale; CTRS-ADHD = ADHD Index; CTRS-E = Emotion Lability Scale); CGI-S = Clinical Global Impression—Severity scale; *p* = Chi-squared test (categorical variables) or Wilcoxon test (continuous variables); OR = odds ratio; CI = confidence interval; Logistic = OR and CI from the multivariate logistic regression model with stepwise type selectio; (*) = *p* < 0.05.

**Table 3 children-08-00598-t003:** Symptom severity perceptions by parents and teachers.

Conners’ Rating Scales	Score	CPRS	CTRS	*p*	K (IC 95%)	Agreement %
Subscales		4909	4909			
O	Pathological	1.938 (39.5)	1.884 (38.4)	0.2637	0.32 (0.29–0.35)	68
	Normal	2.971 (60.5)	3.025 (61.6)			
I	Pathological	3.270 (66.6)	2.682 (54.6)	<0.0001 *	0.25 (0.22–0.28)	64
	Normal	1.639 (33.4)	2.227 (45.4)			
H	Pathological	2.646 (53.9)	2.846 (58.0)	<0.0001 *	0.31 (0.28–0.34)	66
	Normal	2.263 (46.1)	2.063 (42.0)			
ADHD Index	Pathological	3.530 (71.9)	3.478 (70.8)	0.2456	0.26 (0.23–0.29)	70
	Normal	1.379 (28.1)	1.431 (29.2)			
E	Pathological	1.645 (33.5)	1.942 (39.6)	<0.0001 *	0.26 (0.23–0.29)	65
	Normal	3.264 (66.5)	2.967 (60.4)			
CBCL	Score	Mother	Father	*p*	K (IC 95%)	Agreement %
		1082	1082			
I	Pathological	223 (20.6)	159 (14.7)	0.0003	0.64 (0.58–0.70)	89
	Normal	859 (79.4)	923 (85.3)			
E	Pathological	254 (23.5)	204 (18.9)	0.0085	0.69 (0.63–0.74)	89
	Normal	828 (76.5)	878 (81.1)			
T	Pathological	349 (32.3)	255 (23.6)	<0.0001	0.66 (0.61–0.71	86
	Normal	733 (67.7)	827 (76.4)			

CPRS = Conners’ Parent Rating Scale (CPRS-O = Oppositive Scale; CPRS-I = Inattention Scale; CPRS-H = Hyperactivity Scale; CPRS-ADHD = ADHD Index; CPRS-E = Emotion Lability Scale); CTRS = Conners’ Teacher Rating Scale (CTRS-O = Oppositive Scale; CTRS-I = Inattention Scale; CTRS-H = Hyperactivity Scale; CTRS-ADHD = ADHD Index; CTRS-E = Emotion Lability Scale); CBCL = Child Behaviour Checklist (CBCL-I = Internalising problems; CBCL-E = Externalising problems; CBCL-T = Total); *p* = Chi-squared test; K = Cohen’s Kappa Statistic; % = proportion of patients with the same results on tests; (*) = *p* < 0.05.

## Data Availability

The datasets analyzed during the current study are available from the corresponding author upon reasonable request.

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
