# Peer review of "Ten Years (2011–2021) of the Italian Lombardy ADHD Register for the Diagnosis and Treatment of Children and Adolescents with ADHD"

_children, 2021, doi:10.3390/children8070598_

Round 1

Reviewer 1 Report

I have a few doubts and suggestions that should be addressed before publication. The main concerns are related to data interpretation and description. Overall, data are interesting and worth to be published but the way they are presented and discussed should be strongly improved.

  1. Not all methods were described in the Method section (eg.: structure of the figure from the Continuity of Care and Management subtitle)
  2. Tables should be more informative and self-descriptive - all abbreviations should be explained, all statistical methods should be named. It is not clear to what analysis OR or p level refer to, what is comparing?
  3. In the results authors indicate the higher rate of comorbidities. Among them the higher rate of anxiety disorders and conduct disorder. But OR=0.70 and OR=0.49 suggest rather the lower rate of disorders in the ADHD group. Compare with data in the table with %. Authors should carefully check all the data and their interpretation again before the publication.
  4. May authors add the analysis of changes of examined characteristics between 2011 and 2021?
  5. Discussion should be expanded. In its recent form it does not present valuable interpretation of the data based on the comparison of the project results with other publications.
  6. May authors provide the information on limitation of their approach?

Author Response

REVIEWER 1

I have a few doubts and suggestions that should be addressed before publication. The main concerns are related to data interpretation and description. Overall, data are interesting and worth to be published but the way they are presented and discussed should be strongly improved.

  1. Not all methods were described in the Method section (eg.: structure of the figure from the Continuity of Care and Management subtitle)

Response: we added the description of the Radar Chart both in the methods and in the results.

  1. Tables should be more informative and self-descriptive - all abbreviations should be explained, all statistical methods should be named. It is not clear to what analysis OR or p level refer to, what is comparing?

Response: all these information were reported in the data analysis, however we added some notes in the tables.

  1. In the results authors indicate the higher rate of comorbidities. Among them the higher rate of anxiety disorders and conduct disorder. But OR=0.70 and OR=0.49 suggest rather the lower rate of disorders in the ADHD group. Compare with data in the table with %. Authors should carefully check all the data and their interpretation again before the publication.

Response: we corrected these data

  1. May authors add the analysis of changes of examined characteristics between 2011 and 2021?

Response: the aim of the study was to analyse the global 10 years data, anyway the characteristics data between 2011 and 2021 would be too small to draw significant conclusions. 

  1. Discussion should be expanded. In its recent form it does not present valuable interpretation of the data based on the comparison of the project results with other publications.

Response: the aim of the study was to describe the progression of the project and the characteristics of the patients added to the register. The unicity of the project doesn’t allow to compare the results with other publications, except for the ADHD patients general characteristics (e.g. gender distribution, diagnosis, comorbidities, treatments..).

  1. May authors provide the information on limitation of their approach?

Response: we integrated the discussion with the limitation of the study

Reviewer 2 Report

„A review of patient medical records was the design of this study.“ This is not a study design. This is obviously a retrospective study based on medical records.

Figure 1: It does not make sense to compare absolute numbers in age distributions in samples with such imbalances. Histograms with relative frequencies have to be displayed for this purpose.

The stepwise selection process of the logistic regression has to be described in detail and justified with citations as this is very much criticised in the methodological literature.

What do the asterisks in Table 1 mean: <0.0001 *?

Multiple tests are performed which cause alpha inflation. Bonferroni correction has to be applied.

Table 1: Chi-square tests: effect size Cramér V has to be added and discussed as it is well-known that large sample sizes can produce spurious significant results. The same holds for t-tests (effect size Cohen´s d has to be added and discussed).

Table 1 is not comprehensible. OR is the OR resulting from the simple contingency table? Which variables remained as predictors in each logistic regression analysis? A pseudo-R-square (McFadden´s R-square) has to be calculated in order to assess model fit.

Table 2: Same comments as for Table 1. How did you perform logistic regressions on some of the scales (e.g. CGI-S)? Did you dichotimize into pathological/non-pathological? Criteria for this? This is unclear and has to be described in the Methods section. Apart from this, the applied scales have to be described in more detail (reliability, validity) in the Methods section.

Table 3: kappa values <.40 have to be interpreted as just fair. Therefore, kappa corrects by chance agreement and consequently agreement in % should not be mainly looked at.

Where are the explicit analyses refering to the 12-18 month follow-up period mentioned in the abstract?

Author Response

REVIEWER 2

“A review of patient medical records was the design of this study.“ This is not a study design. This is obviously a retrospective study based on medical records.

Response: we provided to modify the sentence

Figure 1: It does not make sense to compare absolute numbers in age distributions in samples with such imbalances. Histograms with relative frequencies have to be displayed for this purpose.

Response: Below the histogram of the F1.

The stepwise selection process of the logistic regression has to be described in detail and justified with citations as this is very much criticised in the methodological literature.

Response: Table 1 stepwise selection

Table 2 stepwise selection

What do the asterisks in Table 1 mean: <0.0001 *?

Response: * = p < 0,05

Multiple tests are performed which cause alpha inflation. Bonferroni correction has to be applied.

Response: we don’t make multiple comparisons tests. In Table 1 we compare ADHD yes and ADHD no. In Table 2 we compare pharmacological vs psychological. In Table 3 CPRS vs CTRS and CBLC mother vs CBCL father

Table 1: Chi-square tests: effect size Cramér V has to be added and discussed as it is well-known that large sample sizes can produce spurious significant results. The same holds for t-tests (effect size Cohen´s d has to be added and discussed).

Response: Below the Cramer V in the tables. For the continuous variables we used the Wilcoxon test, for this reason the Wilcoxon effect size was reported.

Cramer V

ADHD yes

ADHD no

Total

p

Wilcoxon

effect size  (Z/√Z)

V Cramer

Children

4,091

1,843

5,934

Age at diagnosis

median (q1 - q3)

9,0 (7,0 - 11,0)

9,0 (8,0 - 11,0)

9,0 (7,0 - 11,0)

<0.0001  *

4.4705/√5890=0.0582

media (ds)

9,2 (2,6)

9,4 (2,4)

9,2 (2,5)

(min - max)

(5,0 - 17,0)

(5,0 - 17,0)

(5,0 - 17,0)

Missing

30

14

44

School age at diagnosis

5-11

3.264 (80.4)

1.447 (79.1)

4.711 (80.0)

0,2635

0.0146

12-17

797 (19.6)

382 (20.9)

1.179 (20.0)

Missing

30

14

44

Gender

Female

607 (14.8)

367 (19.9)

974 (16.4)

<0.0001  *

0.0634

Male

3.484 (85.2)

1.476 (80.1)

4.960 (83.6)

Missing

-

-

-

Only child

Yes

1.054 (25.8)

396 (21.5)

1.450 (24.5)

0,0004

0.0461

No

3.028 (74.2)

1.443 (78.5)

4.471 (75.5)

Missing

9

4

13

Born in Italy

Yes

3.869 (94.6)

1.774 (96.3)

5.643 (95.1)

0,0067

0.0352

No

220 (5.4)

69 (3.7)

289 (4.9)

Missing

2

-

2

Adopted

Yes

149 (3.6)

34 (1.8)

183 (3.1)

0,0002

0.0481

No

3.938 (96.4)

1.807 (98.2)

5.745 (96.9)

Missing

4

2

6

School

Primary School

3.124 (76.4)

1.371 (74.5)

4.495 (75.8)

0,1124

0.0206

Middle/High School

964 (23.6)

469 (25.5)

1.433 (24.2)

Missing

3

3

6

Support teacher

Yes

514 (12.6)

79 (4.3)

593 (10.0)

<0.0001

0.1277

No

3.577 (87.4)

1.764 (95.7)

5.341 (90.0)

Missing

-

-

-

Educational level of mother

Yes

2.313 (56.5)

1.116 (60.6)

3.429 (57.8)

0,0038

0.0376

No

1.778 (43.5)

727 (39.4)

2.505 (42.2)

Missing

-

-

-

Educational level of father

Yes

1.865 (45.6)

934 (50.7)

2.799 (47.2)

0,0003

0.0472

No

2.226 (54.4)

909 (49.3)

3.135 (52.8)

Missing

-

-

-

Mother employed

Yes

2.729 (66.7)

1.227 (66.6)

3.956 (66.7)

0,9210

0.0013

No

1.362 (33.3)

616 (33.4)

1.978 (33.3)

Missing

-

-

-

Father employed

Yes

3.411 (83.4)

1.624 (88.1)

5.035 (84.9)

<0.0001

0.0612

No

680 (16.6)

219 (11.9)

899 (15.1)

Missing

-

-

-

ADHD familiarity

Yes

831 (20.3)

193 (10.5)

1.024 (17.3)

<0.0001

0.1205

No

3.260 (79.7)

1.650 (89.5)

4.910 (82.7)

Missing

-

-

-

Psychiatric comorbidity

Yes

2.879 (70.4)

1.079 (58.5)

3.958 (66.7)

<0.0001

0.1161

No

1.212 (29.6)

764 (41.5)

1.976 (33.3)

Missing

-

-

-

Type of comorbidity

(flag)

Learning disorder

1.594 (39.0)

613 (33.3)

2.207 (37.2)

<0.0001

0.0546

Sleeping disorder

582 (14.2)

145 (7.9)

727 (12.3)

<0.0001

0.0897

ODD

569 (13.9)

93 (5.0)

662 (11.2)

<0.0001

0.1303

Anxiety

280 (6.8)

162 (8.8)

442 (7.4)

0,0083

0.0343

Language disorder

287 (7.0)

88 (4.8)

375 (6.3)

0,0010

0.0426

Tic

95 (2.3)

22 (1.2)

117 (2.0)

0,0038

0.0376

Conduct disorder

69 (1.7)

41 (2.2)

110 (1.9)

0,1551

0.0185

Coordination disorder

95 (2.3)

23 (1.2)

118 (2.0)

0,0061

0.0356

Chronic disease

Yes

265 (6.5)

122 (6.6)

387 (6.5)

0,8376

0.0027

No
Missing

3.826 (93.5)
-

1.721 (93.4)
-

5.547 (93.5)
-

Type of chronic disease (flag)

Neurological

91 (2.2)

30 (1.6)

121 (2.0)

0,1324

0.0195

Breathing

60 (1.5)

32 (1.7)

92 (1.6)

0,4365

0.0101

Gastrointestinal

18 (0.4)

11 (0.6)

29 (0.5)

0,4227

0.0104

Modello di regressione logistica: diagnosiadhdid(ref='2') = classeetadiagnosiid sessoid figliounicoid natoitaliaid adottivoid gradoscuolaid ripetenteid sostegnoscolasticoidistruzionesuperioremadreid istruzionesuperiorepadreid occupazionemadreid occupazionepadreid adhdfamigliaid partocesareoid prematuroid sottopesoid ritardomotorioid ritardolinguisticoid comorbidita_app comorbidita_sonno comorbidita_opp comorbidita_ansia comorbidita_ritment comorbidita_umore comorbidita_ling comorbidita_tic comorbidita_cond comorbidita_autismo comorbidita_coord comorbidita_altro patologiecronicheid

To assess the model fit we calculated Test della bontà di adattamento di Hosmer e Lemeshow

Test della bontà di adattamento
di Hosmer e Lemeshow

Chi-quadrato

DF

Pr > ChiQuadr

15.3161

8

0.0533

Since p>0.05 there isn’t an evidence of lack of fit.

Table 2. Clinical Characteristics of the ADHD Patients by Treatment Prescription.

Pharmacological Treatment

Psychological

Treatment

Total

p

V di Cramer

734

3,282

4,016

ADHD

Subtype

Combined

586 (79.8)

1.899 (57.9)

2.485 (61.9)

<0.0001  *

0.1752

Inattentive

122 (16.6)

1.096 (33.4)

1.218 (30.3)

Hyperactive

26 (3.5)

287 (8.7)

313 (7.8)

Missing

-

-

-

QI pathologic

Yes

82 (11.4)

79 (2.4)

161 (4.1)

<0.0001  *

0.1755

No

635 (88.6)

3.166 (97.6)

3.801 (95.9)

Missing

17

37

54

CPRS-O

Pathological

384 (63.6)

1.305 (42.6)

1.689 (46.0)

<0.0001  *

0.1563

Normal

220 (36.4)

1.761 (57.4)

1.981 (54.0)

Missing

130

216

346

CTRS-O

Pathological

310 (59.0)

1.219 (41.7)

1.529 (44.3)

<0.0001  *

0.1257

Normal

215 (41.0)

1.707 (58.3)

1.922 (55.7)

Missing

209

356

565

CPRS-I

Pathological

515 (85.3)

2.176 (70.9)

2.691 (73.3)

<0.0001  *

0.1201

Normal

89 (14.7)

892 (29.1)

981 (26.7)

Missing

130

214

344

CTRS-I

Pathological

362 (69.0)

1.717 (58.6)

2.079 (60.2)

<0.0001  *

0.0756

Normal

163 (31.0)

1.211 (41.4)

1.374 (39.8)

Missing

209

354

563

CPRS-H

Pathological

477 (79.0)

1.840 (60.0)

2.317 (63.1)

<0.0001  *

0.1460

Normal

127 (21.0)

1.228 (40.0)

1.355 (36.9)

Missing

130

214

344

CTRS-H

Pathological

403 (76.6)

1.893 (64.7)

2.296 (66.5)

<0.0001  *

0.0911

Normal

123 (23.4)

1.035 (35.3)

1.158 (33.5)

Missing

208

354

562

CPRS-ADHD

Pathological

557 (92.2)

2.371 (77.3)

2.928 (79.7)

<0.0001  *

0.1378

Normal

47 (7.8)

697 (22.7)

744 (20.3)

Missing

130

214

344

CTRS-ADHD

Pathological

459 (87.3)

2.285 (78.1)

2.744 (79.5)

<0.0001  *

0.0818

Normal

67 (12.7)

642 (21.9)

709 (20.5)

Missing

208

355

563

CGI-S

5-7

516 (71.9)

597 (18.8)

1.113 (28.6)

<0.0001  *

0.4553

1-4

202 (28.1)

2.575 (81.2)

2.777 (71.4)

Missing

16

110

126

Psychiatric comorbidity

Yes

612 (83.4)

2.215 (67.5)

2.827 (70.4)

<0.0001  *

0.1345

No

122 (16.6)

1.067 (32.5)

1.189 (29.6)

Missing

-

-

-

Type of comorbidity

(flag)

Learning Disorder

265 (36.1)

1.296 (39.5)

1.561 (38.9)

0,0890

0.0268

Sleeping disorder

130 (17.7)

443 (13.5)

573 (14.3)

0.0032   *

0.0466

ODD

209 (28.5)

356 (10.8)

565 (14.1)

<0.0001  *

0.1959

Anxiety

70 (9.5)

208 (6.3)

278 (6.9)

0.0020   *

0.0487

Intellectual disability

94 (12.8)

150 (4.6)

244 (6.1)

<0.0001  *

0.1332

Mood disorder

57 (7.8)

170 (5.2)

227 (5.7)

0.0061   *

0.0433

Language disorder

61 (8.3)

221 (6.7)

282 (7.0)

0,1307

0.0239

Tic

40 (5.4)

54 (1.6)

94 (2.3)

<0.0001  *

0.0972

Conduct disorder

29 (4.0)

39 (1.2)

68 (1.7)

<0.0001  *

0.0828

Autism

60 (8.2)

65 (2.0)

125 (3.1)

<0.0001  *

0.1378

Coordination disorder

28 (3.8)

63 (1.9)

91 (2.3)

0.0018   *

0.0492

Other

17 (2.3)

66 (2.0)

83 (2.1)

0,5994

0.0083

Modello di regressione logistica: prescrizionefV1id(ref='2') = sottotipoadhdid qipatologicoid cprsaid ctrsaid cprsbid ctrsbid cprscid ctrscid cprshid ctrshid cprsjid ctrsjid cgisid

comorbidita_app comorbidita_sonno comorbidita_opp comorbidita_ansia comorbidita_ritment comorbidita_umore comorbidita_ling comorbidita_tic comorbidita_cond

comorbidita_autismo comorbidita_coord comorbidita_altro

Test della bontà di adattamento
di Hosmer e Lemeshow

Chi-quadrato

DF

Pr > ChiQuadr

7.2808

3

0.0635

Since p>0.05 there isn’t an evidence of lack of fit.

Table 1 is not comprehensible. OR is the OR resulting from the simple contingency table? Which variables remained as predictors in each logistic regression analysis? A pseudo-R-square (McFadden´s R-square) has to be calculated in order to assess model fit.

Response:  OR is the OR resulting from the simple contingency table. As reported in the method, logistic is the results of the multivariate logistic regression analysis with stepwise selection

Modello di regressione logistica: diagnosiadhdid(ref='2') = classeetadiagnosiid sessoid figliounicoid natoitaliaid adottivoid gradoscuolaid ripetenteid sostegnoscolasticoidistruzionesuperioremadreid istruzionesuperiorepadreid occupazionemadreid occupazionepadreid adhdfamigliaid partocesareoid prematuroid sottopesoid ritardomotorioid ritardolinguisticoid comorbidita_app comorbidita_sonno comorbidita_opp comorbidita_ansia comorbidita_ritment comorbidita_umore comorbidita_ling comorbidita_tic comorbidita_cond comorbidita_autismo comorbidita_coord comorbidita_altro patologiecronicheid

To assess the model fit we calculated Test della bontà di adattamento
di Hosmer e Lemeshow

Test della bontà di adattamento
di Hosmer e Lemeshow

Chi-quadrato

DF

Pr > ChiQuadr

15.3161

8

0.0533

Since p>0.05, there isn’t an evidence of lack of fit.

Table 2: Same comments as for Table 1. How did you perform logistic regressions on some of the scales (e.g. CGI-S)? Did you dichotimize into pathological/non-pathological? Criteria for this? This is unclear and has to be described in the Methods section.

Response: we attached the results of the Cramer V; The Clinical Global Impression – Severity scale (CGI-S) is a 7-point scale that requires the clinician to rate the severity of the patient's illness at the time of assessment, relative to the clinician's past experience with patients who have the same diagnosis. As reported in the manual for psychopharmacology (Guy, W. (1976). ECDEU assessment manual for psychopharmacology. US Department of Health, Education, and Welfare, Public Health Service, Alcohol, Drug Abuse, and Mental Health Administration, National Institute of Mental Health, Psychopharmacology Research Branch, Division of Extramural Research Programs) the range are : CGIS 5-7 = pathological, 1-4 = normal range

Modello di regressione logistica: prescrizionefV1id(ref='2') = sottotipoadhdid qipatologicoid cprsaid ctrsaid cprsbid ctrsbid cprscid ctrscid cprshid ctrshid cprsjid ctrsjid cgisid

comorbidita_app comorbidita_sonno comorbidita_opp comorbidita_ansia comorbidita_ritment comorbidita_umore comorbidita_ling comorbidita_tic comorbidita_cond

comorbidita_autismo comorbidita_coord comorbidita_altro

To assess the model fit we calculated Test della bontà di adattamento
di Hosmer e Lemeshow

Test della bontà di adattamento
di Hosmer e Lemeshow

Chi-quadrato

DF

Pr > ChiQuadr

7.2808

3

0.0635

Since p>0.05, there isn’t an evidence of lack of fit.

Apart from this, the applied scales have to be described in more detail (reliability, validity) in the Methods section.

Response: we added the references

Table 3: kappa values <.40 have to be interpreted as just fair. Therefore, kappa corrects by chance agreement and consequently agreement in % should not be mainly looked at.

Response: we are agree with the comment

Round 2

Reviewer 1 Report

One of my comments was:

  1. Discussion should be expanded. In its recent form it does not present valuable interpretation of the data based on the comparison of the project results with other publications.

I don't agree with the response:

Response: the aim of the study was to describe the progression of the project and the characteristics of the patients added to the register. The unicity of the project doesn’t allow to compare the results with other publications, except for the ADHD patients general characteristics (e.g. gender distribution, diagnosis, comorbidities, treatments..).

The importance of the project is related to the possibility of making a comparisons between the regions of the country or with other countries. So, "patients general characteristics (e.g. gender distribution, diagnosis, comorbidities, treatments" should be discussed in details and compared with other studies  - this should be important part of this article and would increase the value of the study. Are these characteristics similar to the results of previous studies/ regions/countries? 

Author Response

  1. Discussion should be expanded. In its recent form it does not present valuable interpretation of the data based on the comparison of the project results with other publications.

I don't agree with the response:

Response: the aim of the study was to describe the progression of the project and the characteristics of the patients added to the register. The unicity of the project doesn’t allow to compare the results with other publications, except for the ADHD patients general characteristics (e.g. gender distribution, diagnosis, comorbidities, treatments..).

The importance of the project is related to the possibility of making a comparisons between the regions of the country or with other countries. So, "patients general characteristics (e.g. gender distribution, diagnosis, comorbidities, treatments" should be discussed in details and compared with other studies  - this should be important part of this article and would increase the value of the study. Are these characteristics similar to the results of previous studies/ regions/countries? 

Response: The previous response was not corrected although comparisons with other studies were reported.

We apologize and thank the reviewer. In the current revision we have further expanded the discussion by agreeing with what you suggest.

Reviewer 2 Report

The histogram in Figure 1 is still based on absolute numbers. Relative frequencies have to be chosen, otherwise the two samples are not comparable in this variable due to the different sample sizes.

"Children were 9 years old" is too unspecific. "Children had a median age of 9 years" is the correct expression.

The applied questionnaires are still not described in detail as it is supposed to be in a scientific article (reliability, validity).

"we don’t make multiple comparisons tests". The authors of course perform multiple tests. In Table 3 eight p values are listed, obviously originating from 8 Chi-square tests. Therefore, Bonferroni correction has to be p=.05/8= .006. This also has to be done in Tables 1 and 2 and p values have to be interpreted accordingly.

The selection process of logistic regression (forward?, backward?) is still not adequately described and discussed.

Cramér-V and Wilcoxon effect sizes have been added but they have not been interpreted in the text. This is incorrect.

Table 3: The authors have not adapted their interpretation: “substantial” agreement between parents and teachers (>60%)" although kappa values signal just a fair agreement. This is not methodologically sound.

Table 3: Chi-square tests have to be added with effect size Cramér-V and this has to be interpreted due to possible spurious statistical significance due to large sample size.

Author Response

Commento 1

  1. The histogram in Figure 1 is still based on absolute numbers. Relative frequencies have to be chosen, otherwise the two samples are not comparable in this variable due to the different sample sizes.

Response

Below the chart with the relative frequencies of the two sample

Commento 2

"Children were 9 years old" is too unspecific. "Children had a median age of 9 years" is the correct expression.

Response

We modified the text

Commento 3

The applied questionnaires are still not described in detail as it is supposed to be in a scientific article (reliability, validity).

Risposta

The rating scales cited are the most used and approved scales for the assessment of ADHD diagnosis worldwide. The diagnostic assessment pathway was agreed, approved and shared by all participating ADHD centres. For additional information about psychometrics properties, validity and reliability, we refer to

Goyette, C.H.; Conners, C.K.; Ulrich, R.F. Normative data on revised Conners’ Parent and Teacher Rating Scales. J. Abnorm. Child Psychol. 1978 6, 221-236

Conners, C.K.; Sitarenios, G.; Parker, J.D.; Epstein, J.N. The revised Conners’ Parent Rating Scale (CPRS-R): Factor structure, reliability, and criterion validity. J. Abnorm. Child Psychol. 1998, 26, 257-268

Achenbach, T. M.; Edelbrock C.S.. Manual for the child behavior checklist and revised child behavior profile. 1983, 85-121.

Guy, W. Clinical Global Impressions. ECDEU Assessment Manual for Psychopharmacology—Revised. 1976; pp. 218-222 Rockville, USA

Commento 4

"we don’t make multiple comparisons tests". The authors of course perform multiple tests. In Table 3 eight p values are listed, obviously originating from 8 Chi-square tests. Therefore, Bonferroni correction has to be p=.05/8= .006. This also has to be done in Tables 1 and 2 and p values have to be interpreted accordingly.

Risposta

These are bivariate analysis all done with independent variables, where we don’t make pairwise comparisons, so in our opinion, the Bonferroni correction for the p-value significance is not applicable.

Commento 5

The selection process of logistic regression (forward?, backward?) is still not adequately described and discussed.

Risposta

The two models are calculated with stepwise selection method: the SELECTION=STEPWISE option is similar to the SELECTION=FORWARD option except that effects already in the model do not necessarily remain. Effects are entered into and removed from the model in such a way that each forward selection step can be followed by one or more backward elimination steps. The stepwise selection process terminates if no further effect can be added to the model or if the current model is identical to a previously visited model.

These are the selection processes for the models in Tables 1 and 2:

Table 1 – Selection process

Step

Added effect

Removed effect

DF

Chi-square

p

1

comorbidita_opp

 -

1

102.5897

<0.0001

2

adhdfamigliaid

 -

1

81.3227

<0.0001

3

sostegnoscolasticoid

 -

1

84.3015

<0.0001

4

comorbidita_sonno

 -

1

34.4831

<0.0001

5

comorbidita_app

 -

1

22.9037

<0.0001

6

sessoid

 -

1

14.6483

0.0001

7

occupazionepadreid

 -

1

13.9731

0.0002

8

comorbidita_ansia

 -

1

10.6092

0.0011

9

comorbidita_cond

 -

1

10.4426

0.0012

10

figliounicoid

 -

1

10.2340

0.0014

11

comorbidita_tic

 -

1

6.9213

0.0085

12

comorbidita_ling

 -

1

6.9515

0.0084

13

comorbidita_coord

 -

1

4.7367

0.0295

14

adottivoid

 -

1

3.9338

0.0473

15

occupazionemadreid

 -

1

4.0118

0.0452

16

gradoscuolaid

 -

1

3.9069

0.0481

Table 1 – Type III effects analysis (significance of an effect with all the other effects in the model)

Effect

DF

Wald Chi-square

p

sessoid

1

15.8653

<0.0001

figliounicoid

1

8.1997

0.0042

adottivoid

1

5.1050

0.0239

gradoscuolaid

1

3.9040

0.0482

sostegnoscolasticoid

1

63.6452

<0.0001

occupazionemadreid

1

4.2626

0.0390

occupazionepadreid

1

7.5892

0.0059

adhdfamigliaid

1

81.6339

<0.0001

comorbidita_app

1

25.9422

<0.0001

comorbidita_sonno

1

35.6652

<0.0001

comorbidita_opp

1

76.0304

<0.0001

comorbidita_ansia

1

10.8090

0.0010

comorbidita_ling

1

5.4373

0.0197

comorbidita_tic

1

6.5092

0.0107

comorbidita_cond

1

10.3260

0.0013

comorbidita_coord

1

4.4485

0.0349

Table 1 – Hosmer-Lemeshow lack of fit test: p=0.0533

Table 2 – Selection process

Step

Added effect

Removed effect

DF

Chi-square

p

1

cgisid

 -

1

555.4480

<0.0001

2

cprsHid

 -

1

42.4792

<0.0001

3

sottotipoadhdid

 -

2

33.2481

<0.0001

4

comorbidita_tic

 -

1

19.1685

<0.0001

5

comorbidita_ritment

 -

1

18.8364

<0.0001

6

comorbidita_coord

 -

1

15.9910

<0.0001

7

comorbidita_opp

 -

1

14.2897

0.0002

8

cprsCid

 -

1

6.7926

0.0092

Table 2 – Type III effects analysis (significance of an effect with all the other effects in the model)

Effect

DF

Wald Chi-square

p

sottotipoadhdid

2

20.0618

<0.0001

cprsCid

1

6.7527

0.0094

cprsHid

1

22.2341

<0.0001

cgisid

1

298.2738

<0.0001

comorbidita_opp

1

12.9858

0.0003

comorbidita_ritment

1

19.3065

<0.0001

comorbidita_tic

1

18.8708

<0.0001

comorbidita_coord

1

18.0991

<0.0001

Table 2 – Hosmer-Lemeshow lack of fit test: p=0.0635

Commento 6

Cramér-V and Wilcoxon effect sizes have been added but they have not been interpreted in the text. This is incorrect.

Risposta:

We added the interpretation in the methods in the text.

Commento 7

Table 3: The authors have not adapted their interpretation: “substantial” agreement between parents and teachers (>60%)" although kappa values signal just a fair agreement. This is not methodologically sound.

Risposta:

We adapted the interpretation.

Commento 8

Table 3: Chi-square tests have to be added with effect size Cramér-V and this has to be interpreted due to possible spurious statistical significance due to large sample size.

Risposta:

We added V Cramer effect size

Conners’ Rating Scales

Score

CPRS

CTRS

p

V

K (IC 95%)

Agreement %

Subscales

4,909

4,909

O

Pathological

1.938 (39.5)

1.884 (38.4)

0,2637

0.01

0.32 (0.29 - 0.35)

68

Normal

2.971 (60.5)

3.025 (61.6)

I

Pathological

3.270 (66.6)

2.682 (54.6)

<0.0001

0.12

0.25 (0.22 - 0.28)

64

Normal

1.639 (33.4)

2.227 (45.4)

H

Pathological

2.646 (53.9)

2.846 (58.0)

<0.0001

0.04

0.31 (0.28 - 0.34)

66

Normal

2.263 (46.1)

2.063 (42.0)

ADHD Index

Pathological

3.530 (71.9)

3.478 (70.8)

0,2456

0.01

0.26 (0.23 - 0.29)

70

Normal

1.379 (28.1)

1.431 (29.2)

E

Pathological

1.645 (33.5)

1.942 (39.6)

<0.0001

0.06

0.26 (0.23 - 0.29)

65

Normal

3.264 (66.5)

2.967 (60.4)

CBCL

Score

Mother

Father

p

V

K (IC 95%)

Agreement %

1,082

1,082

I

Pathological

223 (20.6)

159 (14.7)

0.0003

0.08

0.64  (0.58 - 0.70)

89

Normal

859 (79.4)

923 (85.3)

E

Pathological

254 (23.5)

204 (18.9)

0.0085

0.06

0.69 (0.63 - 0.74)

89

Normal

828 (76.5)

878 (81.1)

T

Pathological

349 (32.3)

255 (23.6)

< 0.0001

0.10

0.66 (0.61 - 0.71

86

Normal

733 (67.7)

827 (76.4)